# Alteration of Mitochondrial Transcript Expression in *Arabidopsis thaliana* Using a Custom-Made Library of Pentatricopeptide Repeat Proteins

**DOI:** 10.3390/ijms241713233

**Published:** 2023-08-26

**Authors:** Lilian Vincis Pereira Sanglard, Ian D. Small, Catherine Colas des Francs-Small

**Affiliations:** Australian Research Council Centre of Excellence in Plant Energy Biology, School of Molecular Sciences, The University of Western Australia, Crawley, WA 6009, Australia

**Keywords:** plant mitochondria, pentatricopeptide repeat (PPR) proteins, ATP synthase, RNA processing factor 2 (RPF2), biotechnology

## Abstract

Pentatricopeptide repeat (PPR) proteins are considered a potential tool for manipulating organelle gene expression in plants because they can recognise a wide range of different RNA sequences, and the molecular basis for this sequence recognition is partially known and understood. A library of redesigned PPR proteins related to restorer-of-fertility proteins was created and transformed into plants in order to target mitochondrial transcripts. Ninety different variants tested in vivo showed a wide range of phenotypes. One of these lines, which displayed slow growth and downward curled leaves, showed a clear reduction in complex V. The phenotype was due to a specific cleavage of *atp1* transcripts induced by a modified PPR protein from the library, validating the use of this library as a source of mitochondrial ‘mutants’. This study is a step towards developing specific RNA targeting tools using PPR proteins that can be aimed at desired targets.

## 1. Introduction

Mitochondria are one of the most important cellular organelles, playing a key role in cellular respiration in plants. Mitochondrial genes mostly encode membrane proteins of the respiratory chain, i.e., *nad1* to *nad4*, *nad4L*, *nad5* to *nad7*, *nad9* (complex I) [1,2,3,4,5], *cob* (complex III) [4,6], *cox1* to *cox3* (complex IV) [7], *atp1*, *atp4*, *atp6*, *atp8* and *atp9* (complex V) [4,8]. In addition, they encode proteins related to cytochrome *c* biogenesis, ribosomal subunits, protein translocation, RNA splicing as well as rRNAs and tRNAs. Many of these genes encode subunits of very large protein complexes and it is unclear how the biogenesis, assembly and activity of these complexes are controlled. One hypothesis suggests the control is led by genes that are retained within the mitochondrial genome [9]. Mitochondrial gene expression can be regulated at different levels, such as transcriptional, post-transcriptional, translational and post-translational [10]. Many RNA-binding proteins participate in these processes such as the CRM-domain protein mCSF1 [11], the PORR-domain protein WTF9 [12], mTERF factors [13] and the large family of pentatricopeptide repeat (PPR) proteins [14]. PPR proteins are involved in several stages of post-transcriptional RNA modifications such as intron splicing, RNA editing, stability and processing [15,16]. Each PPR motif recognises and binds a single nucleotide with a 1:1 correspondence between contiguous protein motifs and RNA bases in the target RNA [17,18,19,20]. This recognition is determined primarily by hydrogen bonding patterns between the RNA base and two amino acid side chains at positions 5 and 35 in each PPR motif [17,20]. According to the PPR code proposed by Barkan and Small [15], the most common P-class PPR motif combinations at positions 5 and 35 are asparagine and aspartic acid (ND), asparagine and asparagine (NN), serine and aspartic acid (SD) and serine and asparagine (SN), corresponding, respectively, to the nucleotides uracil (U), cytosine (C), guanine (G) and adenine (A) in the RNA target [15].

Cytoplasmic male sterility (CMS) is one of the most studied nuclear–mitochondrial interactions, causing failure to generate viable pollen [21]. PPR proteins responsible for suppressing CMS belong to the RF (restorer-of-fertility) subgroup and their close homologues in *Arabidopsis thaliana* are known as RFL (RF-like) proteins. Most RF proteins restore pollen fertility by promoting cleavage of CMS-inducing mitochondrial transcripts [22,23]. Rf-PPR592 from Petunia was the first RF protein identified to restore pollen fertility by reducing the amount of PCF (Petunia CMS-associated fused) protein in mitochondria that could lead to CMS [24,25]. Many plants contain RFL genes even though CMS is rare or absent in the species and most RFL proteins appear not to be associated with CMS suppression. Instead, they have been linked to the 5′-end-processing of mitochondrial transcripts [26,27]. RF proteins show a high degree of natural variability due to positive selection for diversity at the positions determining RNA sequence recognition [28]. Therefore, this subgroup of PPR proteins is a logical choice for the rational engineering of new RNA processing tools. The PPR code, although not fully understood, opens the possibility to alter their target specificity by modifying the amino acids at key positions for RNA recognition and binding.

RPF2 (RNA PROCESSING FACTOR2) was the first RFL gene whose function was characterized in Arabidopsis and is found in a large cluster of related RFL genes on chromosome 1. It has two known natural targets, located within the 5′ untranslated regions of *cox3* and *nad9* (Appendix A) [26]. We previously demonstrated that it is possible to use RPF2 proteins to knockdown expression of mitochondrial-targeted transcripts [29,30]. In this work, a library of such RF proteins was generated in order to produce a collection of Arabidopsis lines altered in the expression of various mitochondrial transcripts. We describe the creation, validation and high-throughput screening strategy of this library as well as the characterization of a line altered in ATP synthase, and discuss the potential uses of such a library to study mitochondrial function, physiology and complex assembly.

## 2. Results

### 2.1. Creating a Library of RPF2 Proteins

The aim in this work was to use RPF2 as the base for a library of PPR variants capable of targeting numerous mitochondrial targets. However, according to the PPR code, the RPF2 sequence contains non-standard amino acid–RNA base combinations at positions 5 and 35 in the P6 (GG) and P9 (SS) motifs, which align with A nucleotides (Appendix A). These motifs were modified via mutagenesis to produce the more canonical sequence referred to as ‘mRPF2’. The codon encoding G at position 5 in the P6 motif was altered to encode S, and the codon encoding G at position 35 was altered to encode N. In the P9 motif, the codon encoding S at position 35 was also altered to encode N. In order to check if the modifications introduced in the mRPF2 sequence would interfere with RNA binding or specificity in vitro, both native RPF2 and mRPF2 were expressed in bacteria and their binding to the RPF2 natural target *cox3* was tested via an RNA Electrophoretic Mobility Shift Assay. Figure 1 and Appendix A show that mRPF2 binds to *cox3* in vitro in the same way as the native RPF2.

In order to check this result in vivo, the same modifications were introduced via PCR into the RPF2-*nad6* construct, a version of RPF2 targeting the *nad6* transcript [29], which was then used to transform Arabidopsis via *Agrobacterium tumefaciens*. As expected for a complex I mutant, the mRPF2-*nad6* and original RPF2-*nad6* plants had similar phenotypes, i.e., slow growth (Figure 2) and undetectable complex I (CI) (Figure 3). These results allowed us to use mRPF2 as the scaffold for the library design.

In order to generate the PPR library, overlapping gBlocks Gene Fragments (IDT) encoding variant mRPF2 motifs were optimised for Gibson assembly. Five fragments were created using the 16 mRPF2 motifs. Each fragment comprised four PPR motifs: two outside fixed motifs (P1–P4, P4–P7, P7–P10, P10–P13, P13–P16) surrounding two internal motifs altered either at position 35 (P2, P5, P8, P11, P14) or position 5 (P3, P6, P9, P12, P15), yielding a theoretical total of 1024 possible variants (Figure 4 and Appendix A). In total, 90 distinct constructs were obtained out of 384 clones sequenced. Potential mitochondrial targets for these 90 sequences (Appendix A) were predicted using the frequency tables from Kobayashi et al., 2019 [31].

### 2.2. Genotype and Phenotype Analysis

After transfer of the 90 constructs into the pCAMBIA1380 binary vector and plant transformation, 64 different variant Arabidopsis lines out of 177 T2 transformants were identified via a qPCR SNP method (Appendix A). Variant constructs were sorted by using two allele-discriminating primers, one for detecting the WT sequence, and another for identifying the sequence modified in each motif. An additional mismatch was incorporated at the adjacent (penultimate) site of each primer to increase the specificity. The phenotypes were visually inspected on T2 plates containing 10–14 seedlings for each line as shown in Figure 5a. The native RPF2 (WT + RPF2) line [29] was used as a control. Two to eleven independent transformants were obtained for most constructs, but for twenty-two variant constructs, only one transformant could be isolated. Most of the independent transformants bearing the same construct had similar phenotypes. Eighty-two transformants had clearly distinct phenotypes from the WT + RPF2 control (Appendix A) whilst the other ninety-five lines were similar to WT + RPF2 (Appendix A). Transformants were classified by phenotype (Appendix A). Some phenotypes were slow growth, severe growth defect, dark green leaves and purple hypocotyls, and green vitreous leaves or curved leaves (Appendix A). To give an overview of the phenotype diversity, one seedling of each line is shown in Appendix A.

To evaluate the growth rates of the transformants on the plates, the mean leaf area per seedling was estimated via image analysis at 2 and 3 weeks (Figure 5). The graphs (Figure 5b) show a bimodal distribution of seedling growth, with 50/192 transformed lines (including #9 and #280) grouped with the slow-growing transformant RPF2-*nad6* and the remaining two thirds grouped with the fast-growing WT + RPF2 control.

### 2.3. Analysis of the Mitochondrial Respiratory Complexes

Most T2 transformants (177 out of 192) were screened through a simplified analysis of the mitochondrial respiratory complexes via blue native polyacrylamide gel electrophoresis (BN-PAGE) and Western blotting (WB). Considering the number of lines to screen and the small amount of tissue available in early generations, a crude membrane extraction protocol only requiring 100 mg of fresh tissue was used [12,32]. Wild types and the Nad6-deficient RPF2 variant (RPF2-*nad6*) mostly lacking assembled complex I [29] were used as controls on each gel. After transfer, the membranes were incubated with a mixture of anti-Nad9 (complex I) and anti-Rieske protein (complex III) antibodies, followed by incubation with an anti-Atp1 (complex V) antibody, to provide a quick overview of the respiratory complexes in the different lines. Unfortunately, this method was not quantitative enough to reliably detect small variations in mitochondrial complex abundances (Appendix A), due to variability in protein extraction yield and transfer efficiency. It was nevertheless adequate to detect the quasi-absence of complex V in transformant #280 (variant 65).

### 2.4. Further Characterisation of Transformant #280

In total, 50 out of 66 seeds of the transformant #280 seeds sown on hygromycin plates germinated in the T2 generation. The 24 seedlings transferred to soil were all smaller than WT with downward curled leaves. The seedlings showed delayed development (Figure 6) but were able to produce low amounts of viable seeds. The quasi-absence of assembled complex V detected via BN PAGE suggested a deficiency in one of the ATPase subunits and the predicted target for construct #65 was indeed *atp1* (Appendix A). Mitochondrial transcript analysis via qRT-PCR confirmed that *atp1* levels were reduced in all three #280 plants analysed (Figure 7) whilst *atp6-1*, *atp6-2* and *atp8* were slightly overexpressed.

In order to check whether the *atp1* transcript was cleaved in transformant #280, Northern blots were hybridized with the antisense oligonucleotide probe 1457-AS localized near the 3′ end of the *atp1* transcript. In WT, the 1457-AS probe hybridized to a single transcript corresponding to the expected size (2045 nt) of the full mature *atp1* transcript. In transformant #280 plants, this hybridization signal was weak or absent, indicating a loss of full transcript, but all transformants showed a strong signal at ~289 nt, indicating a cleavage product (Figure 8a,b). The pattern was identical to that obtained for the RPF2-*atp1* transformants [30]. In the RPF2-*atp1* variant, the specificity of RPF2 was deliberately altered to target *atp1* transcripts, using the PPR code [17]. The codon encoding S at position 5 in the P8 motif was altered to encode N. In the P13 motif, the codon encoding N at position 5 was altered to encode S, and the codon encoding T at position 35 was altered to encode N. In the P15 motif, the codon encoding N at position 35 was also altered to encode D [30]. Transformant #280 has the same modifications as RPF2-*atp1* in motifs 8 and 15 (Appendix A). Although RPF2-*atp1* has an extra modification at motif 13 (Appendix A), circular RT-PCR (cRT-PCR) confirmed that the cleavage happens at the same site as for transformant #280 plants (Figure 8c and Appendix A).

An untargeted proteomics approach was used to estimate the amounts of the various subunits of ATP synthase in the transformed plants, and the results showed a general decrease in the abundance of most of the ATP synthase subunits, except the associated proteins IF1 (inhibitor of ATP synthase activity), e (ATP20) and g (ATP21) (Figure 9, Appendix A), as previously shown for RPF2-*atp1* [30].

## 3. Discussion

We successfully created a library of PPR-RFL proteins, aiming to alter the expression of various mitochondrial transcripts, based on the method we used to specifically inactivate the expression of *nad6* [29] and *atp1* [30]. Nevertheless, scaling up was not straightforward and we met multiple challenges. Although the phenotypic analysis of the PPR variants in planta revealed a wide range of phenotypes, 15 out of 192 transformants were not viable in the T2 generation. Genotyping analysis of the 177 remaining transformants revealed that only 64 out of 90 constructs had been transferred into these plants (Appendix A). Variants may have been lost through competition because the plant transformation was performed in pools of sixteen variants. Most of the 64 variants showed consistent phenotypes between independent transformants; however, we observed different phenotypes for variants 16, 34 and 88 (Appendix A). This suggests that they had either integrated several copies or had higher expression levels of the construct than others, which is quite common after plant transformation [33,34]. We cannot exclude that some plants might have received more than one construct and were therefore expressing more than one mRPF2 variant. The phenotypic analysis of 177 transformants could not identify any visible defects in 95 transformants (Appendix A), which does not exclude that they have RNA processing defects as other mitochondrial mutants affected in RNA processing, such as *rpf1*, *rpf2*, *rpf3*, *rpf4*, *rpf6*, *rfl2* and *rfl9* [26,27,35,36,37,38], do not show any visible growth phenotype. In contrast, a wide range of phenotypes was identified in 82 transformants (Appendix A). Some of these (such as dark green leaves, purple hypocotyl and vitreous leaves) had not been reported in the literature as mitochondrial defects related to RNA processing. The growth rate analysis of the transformants (Figure 5) showed a bimodal distribution of seedling growth, suggesting that the library generated about 50 new “mitochondrial mutants” (including #9 and #280) as suggested by their slow-growth phenotype, a trait very commonly encountered by plants with mitochondrial defects, such as the complex I-deficient transformant RPF2-*nad6.* One important caveat is that the plants were on a hygromycin-containing medium, which can also cause damage and abnormal development [39], potentially explaining the phenotype observed in some lines.

Another challenge was related to establishing high-throughput screening methods to link the phenotypes observed to mitochondrial defects. BN-PAGE of total membrane proteins followed by Western blotting [32] was used initially because this method uses the least amount of tissue to perform a rapid screen. Although this approach is not quantitative enough to detect minor changes in mitochondrial complexes and only covers some of the possible mitochondrial targets, one clear complex V mutant (transformant #280) was identified. Transformant #280 plants were characterized by the absence of assembled complex V (Appendix A), resulting in slow growth and downward curled leaves (Figure 5 and Figure 6). We successfully found the new cleavage site promoted by the modified RPF2-280 in the *atp1* transcript, which happened to be the same as for RPF2-*atp1* as shown via circular RT-PCR (Figure 8c).

The predicted targets for the library of ‘mutants’ developed by modifying PPR proteins include mitochondrially encoded subunits from complex I (CI), complex IV (CIV) and complex V (CV) (Appendix A). Additional predicted targets are transcripts encoding proteins of the cytochrome *c* maturation complex, the translation machinery and protein translocation. Most previous studies focused on the knockdown of a single specific subunit of a mitochondrial complex and its consequences. However, this PPR library potentially provides a series of ‘mutants’, each lacking a different subunit of the same mitochondrial complex. Analysing these ‘mutants’ should help answer questions about the regulation of the biogenesis, assembly and activity of these mitochondrial complexes. Knocking down the expression of different subunits of a complex will be helpful for reconstructing these assembly pathways, helping to understand in which order the subunits are assembled. Knocking down the expression of different subunits of a complex can also answer questions about feedback on the expression of other subunits in the same complex.

Another use of this type of library would be to give insights into the binding specificity of PPR proteins. Here, we have the example of two RPF2 variants binding the *atp1* transcript (this work and RPF2-*atp1* transcript [30]). RPF2-*atp1* and transformant #280 appear to share the same binding and cleavage site although they differ in motif #13. Both RPF2-*atp1* and transformant #280 have S modified to N at position 5 of motif 8 and N modified to D at position 35 of motif 15 (Appendix A). RPF2-*atp1* has two additional modifications in motif 13, N to S at position 5 and T to N at position 35 (Appendix A), that would be predicted to improve the recognition of the *atp1* sequence [17,40]. However, the lack of these modifications in transformant #280 shows that they are not needed for *atp1* binding and cleavage. Therefore, we suspect that the change of SD to ND at motif 8 near the centre of the PPR–RNA interface was the crucial modification that targeted both RPF2-*atp1* and transformant #280, rather than the modifications at motifs 13 and 15.

In conclusion, engineered RNA-binding proteins such as PPR proteins hold plenty of promise for RNA manipulation in organelles, and we successfully demonstrated that modified RPF2 binds and leads to effective cleavage of two specific RNA targets, *nad6* [29] and *atp1* (this work and RPF2-*atp1* transcript [30]). Nevertheless, modifying a natural RFL protein is constrained by the need to alter more than one or two motifs, which could potentially compromise protein stability, thus limiting potential targets to sequences similar to the original binding site. The low flexibility of modified RFLs can be overcome if entirely synthetic RFLs can be designed; there would then be no limitation on target choice. Synthetic PPR proteins have been successfully used as editing factors [41] in plant organelles and in *E. coli* [42]. They are a good tool for studying PPR specificity [43,44].

## 4. Materials and Methods

### 4.1. Plant Growth

Arabidopsis thaliana (Col-0) and transformant plants were grown from seeds on 0.5 × MS solid medium (2.2 g/L Murashige and Skoog (MS) Basal Medium (PhytoTech, Lenexa, KS, USA) containing 0.5 g/L MES, 5 g/L sucrose, pH 5.8, and 4 g/L plant agar. The seeds were sterilised for 5 min in a 70% ethanol and 0.05% Triton X-100 solution, then washed in 100% ethanol and dried in a laminar flow hood. Dried seeds were then sown onto Petri dishes containing 0.5× MS medium, stratified at 4 °C for 48 h and transferred to 22 °C in a controlled growth facility under long-day conditions (16 h of 100 μEm^−2^s^−1^ light, 8 h dark) for two weeks. The seedlings were transferred to soil and grown under long-day conditions.

### 4.2. Construction of the PPR Library

To generate the PPR library, variant RPF2 motifs were designed as gBlocks Gene Fragments (IDT, Coralville, IA, USA), which can vary between 251 and 500 bp in length, with 18 potential variable bases in the central part, and a minimum 125 bp fixed sequence at each end. The PPR fragments (Appendix A, Figure 4) were cloned via Gibson assembly (GA) [45] into the pETM20 vector, previously digested by *XhoI* and *NcoI*, and used to transform *E. coli* (DH5α). The colonies were screened via PCR to check the insert size. Bacterial colonies from the samples with the expected size were used to inoculate 5 mL of LB medium, containing 50 μg/mL carbenicillin and grown O/N at 37 °C with constant shaking at 220 rpm. Cells were pelleted via centrifugation at 4000× *g* for 15 min. Plasmid DNA isolation was performed using the FavorPrep Plasmid Extraction Mini Kit (Favorgen Biotech Corp, Ping Tung, Taiwan). Plasmid DNA was eluted in 30 μL of water and sequenced using Macrogen (Seoul, Republic of Korea). The validated PPR variants were used to transform the C41 strain for bacterial protein expression.

For the preliminary in vitro test of the mRPF2 PPR scaffold, the PPR variants were used to transform the C41 strain for bacterial protein expression. The total protein was separated via SDS-PAGE, transferred to PVDF membrane (Bio-Rad, Hercules, CA, USA) and incubated with an anti-*His* antibody (Sigma SAB4200620, Merck, Rahway, NJ, USA). The proteins were purified using the Nickel-charged resin (Nuvia, BioRad, Hercules, CA, USA). REMSA was performed as described previously [18] using a 5′-Fluorescein-labelled RNA oligonucleotide, *cox3* (Flc-AGGAAAUACGAACCCUCCAA). The gels were imaged with a Typhoon imager (GE Healthcare, Chicago, IL, USA). Fluorescein was excited at 488 nm and detected through a 520 nm band-pass filter.

In order to test the PPR library in vivo, the PPR variants were transferred to the pCAMBIA1380 vector as follows: PCR amplification from the variants in pETM20 was performed using a reverse primer with a 36 bp overlap to the pCAMBIA1380 vector (Appendix A). These PCR products (1478 bp) and two extra gBlocks (one containing the NOS promoter and the *Solanum tuberosum* Formate dehydrogenase (FDH) targeting sequence, and another containing the C-terminal 3-times FLAG tag), as well as the *BamHI* and *EcoRI* double digested pCAMBIA1380 (8861 bp) vector, were assembled via GA [45]. Pools of sixteen PCR products at a time were used for the GA reaction. The pools of variants were transferred to *Agrobacterium tumefaciens* and introduced into Arabidopsis plants via floral dip [46].

### 4.3. Genotypic Analysis

For the identification of the transgene construct in each transformant, twenty specific primers were designed using the SNP method [47]. Each pair of primers was tested to find the ideal annealing temperature using a temperature gradient in the PCR machine (Appendix A). The qRT-PCR assay was performed in a LightCycler480 instrument (Roche Diagnostics, Basel, Switzerland) using the LC480 SYBR Green Master Mix (Takara, Kusatsu shi, Japan) with the following thermal cycling program: 95 °C for 10 min, followed by 45 cycles of 95 °C for 10 s, 60 °C for 10 s and 72 °C for 20 s.

### 4.4. Phenotypic Analysis

The T0 seeds were harvested from transformed plants and sown on MS plates containing 25 μg/mL hygromycin. The T1 seeds were then sown to inspect the phenotypes of the T2 generation and collect samples for further analyses. Photos were taken 2 and 3 weeks after germination on hygromycin plates with a Nikon D60 camera equipped with a Micro-Nikkor 60 mm lens (*f* 5.6; 1/500 s). Leaf area was calculated from an image analysis of photographs of seedlings taken at 2 and 3 weeks of age. The analysis used functions from the JuliaImages packages (https://juliaimages.org (accessed on 12 August 2023)). Images of 100 cm^2^ plates (2278 × 2278 pixels) were normalized to adjust contrast using ImageContrastAdjustment.jl (LinearStretching with default parameters), then the colour difference of each pixel from a reference colour (dark green, #003300) was calculated with the Colors.colordiff function. Pixels differing from ‘dark green’ by less than half of the median difference were considered to be leaf pixels. The total number of such pixels was counted and converted to a mm^2^ value. The number of seedlings on each plate was counted manually (excluding hygromycin-sensitive seedlings) so that the mean leaf area per seedling could be calculated.

### 4.5. High-Throughput BN-PAGE and Western Blotting

BN-PAGE and Western blotting were performed as previously described [32]. A cocktail of anti-Nad9 (complex I) and anti-RISP protein (complex III) antibodies followed by an anti-ATP1 (complex V) antibody were used (Appendix A). The blot was developed with the Clarity Western Blot ECL Substrate (Bio-Rad) using a GE Amersham AI680 imager.

### 4.6. RNA Extraction and qPCR

RNA was extracted using RNAzol (Molecular Research Center, Cincinnati, OH, USA). DNA-free RNA was used for first-strand cDNA synthesis using the SuperScript^®^ III first-strand synthesis system for qRT-PCR (Invitrogen, Waltham, MA, USA). qPCR was performed using the LC480 SYBR Green Master mix (Takara) recommended protocol. Specific primers were used to analyse the expression of all mitochondrial transcripts [48]. Circular RT PCR (cRT-PCR) was performed as previously described [30].

### 4.7. Proteomics Analysis

For quantitative untargeted mass spectrometry, crude mitochondrial pellets were obtained from 3-week-old WT and transformant #280 seedlings grown on plates and prepared as previously described [49,50]. Samples were analysed via LC-MS on a Thermo Exploris 480 mass spectrometer using data-dependent acquisition [30]. 

## Figures and Tables

**Figure 1 ijms-24-13233-f001:**
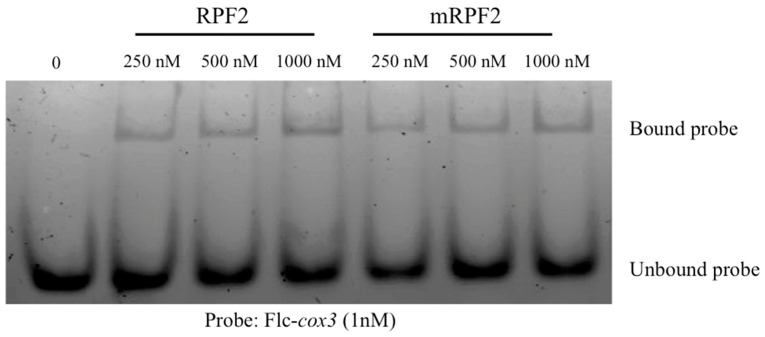
RNA Electrophoretic Mobility Shift Assay showing that both native RPF2 and mRPF2 proteins bind the Flc-*cox3* probe in vitro.

**Figure 2 ijms-24-13233-f002:**
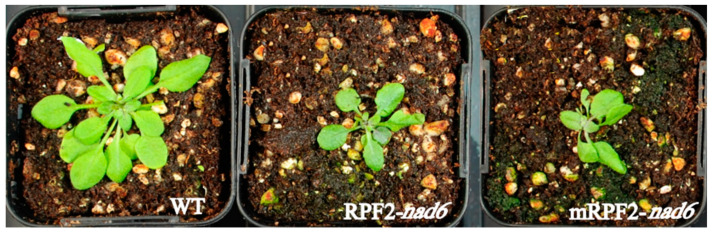
RPF2-*nad6* and mRPF2-*nad6* plants showed a slow-growth phenotype compared to WT (4-week-old rosettes).

**Figure 3 ijms-24-13233-f003:**
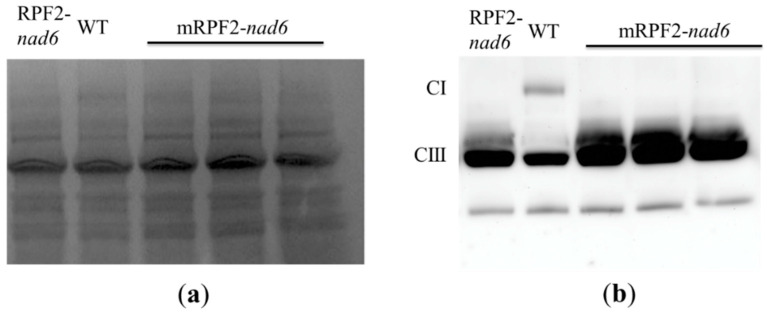
Alteration in complex I (CI) levels in RPF2-*nad6* and 3 independent mRPF2-*nad6* transformants compared to WT: (**a**) separation of mitochondrial inner membrane protein complexes via BN-PAGE; (**b**) a Western blot of the gel probed with anti-Nad9 (complex I) and anti-RISP (complex III) antibodies simultaneously.

**Figure 4 ijms-24-13233-f004:**
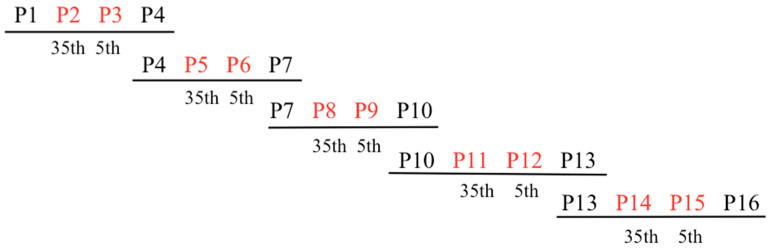
General scheme of gBlocks Gene Fragments covering the 16 mRPF2 PPR motifs. The fixed motifs are in black, the variable motifs in red.

**Figure 5 ijms-24-13233-f005:**
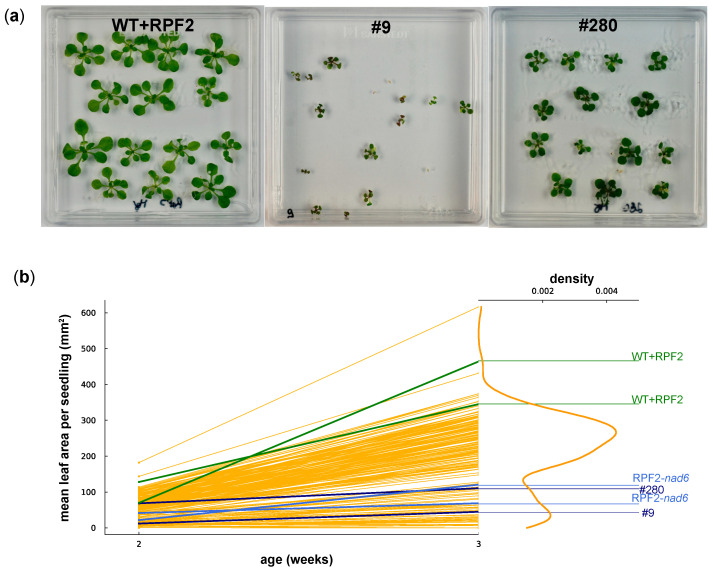
(**a**) WT + RPF2 and transformants #9 and #280 grown on agar plates for 2 weeks. (**b**) Mean leaf area per seedling of 192 transformant lines grown on agar plates at 2 and 3 weeks as compared with native RPF2 (WT + RPF2, green lines) and the RPF2-*nad6* transformant (blue lines). Data for #9 and #280 are shown as navy-blue lines. The right part of the graph shows the density of the distribution of the week 3 leaf area values.

**Figure 6 ijms-24-13233-f006:**
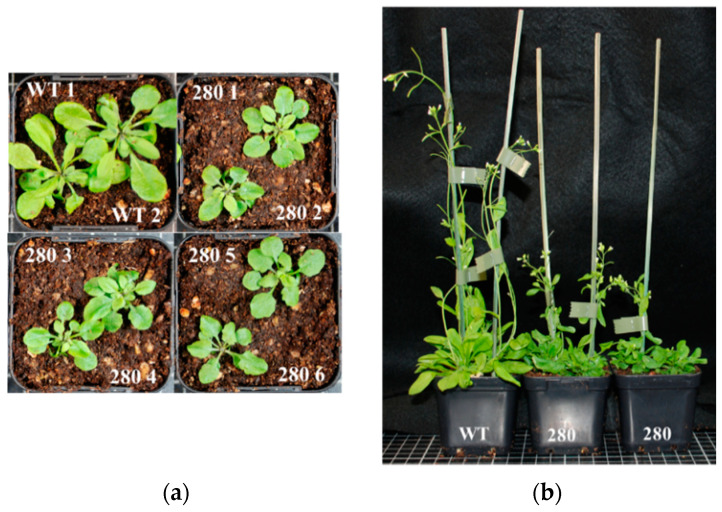
Phenotype of transformant #280 plants compared to WT Col-0: (**a**) four-week-old rosettes; (**b**) six-week-old plants.

**Figure 7 ijms-24-13233-f007:**
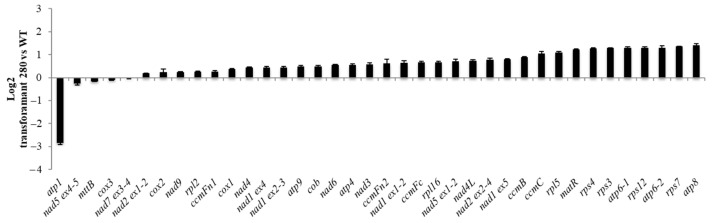
Relative expression of all mitochondrial transcripts in #280 transformant plants compared with WT (Log2 scale). The error bars represent the standard deviation between 3 biological replicates (3 technical replicates each).

**Figure 8 ijms-24-13233-f008:**
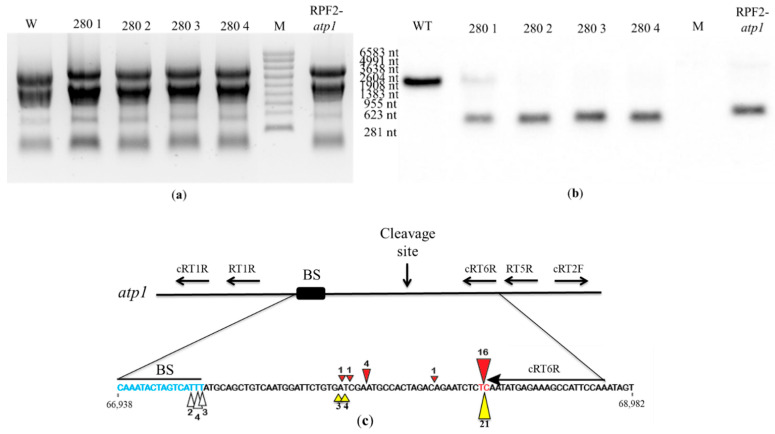
The *atp1* transcript is cleaved in transformant #280 plants: (**a**) RNA gel of WT, transformant #280 and RPF2 plants stained with ethidium bromide. The sizes on the right are molecular weight markers (M). (**b**) A Northern blot of the gel was probed with the *atp1* 1457-AS probe. (**c**) Map of the *atp1* transcript for the uncleaved fragment and 3′ fragment indicating the putative RPF2-*atp1* binding site and the positions of the primers used for the cRT-PCR experiment. The coordinates of this region on the Col-0 mitochondrial genome (BK010421) are 66,938 to 68,982. The predicted binding site (67,277–67,293) and the main cleavage site (67,227–67,228) are highlighted in blue and red, respectively. The red and yellow triangles indicate the 5′ ends of the cleavage products of RPF2-*atp1* and transformant #280, respectively. The figures near the triangles indicate the numbers of clones obtained.

**Figure 9 ijms-24-13233-f009:**
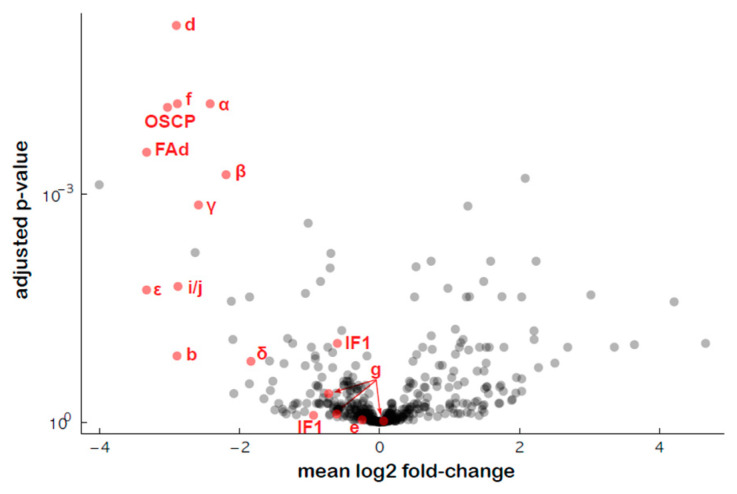
Relative protein abundances of the ATP synthase complex subunits in the #280 transformant estimated via quantitative untargeted mass spectrometry. The *x* axis indicates mean log2 fold difference (WT/#280), the *y* axis indicates the *p*-value for the hypothesis of equal relative abundance, i.e., points towards the top left of the plot are significantly less abundant in #280 samples.

## Data Availability

Quantitative untargeted mass spectrometry data have been deposited to the ProteomeXchange Consortium via the PRIDE [51] partner repository with the dataset identifier PXD037659.

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
