# Peer review of "Alteration of Mitochondrial Transcript Expression in Arabidopsis thaliana Using a Custom-Made Library of Pentatricopeptide Repeat Proteins"

_ijms, 2023, doi:10.3390/ijms241713233_

Round 1
Reviewer 1 Report
The authors describe the use of engineered PPR proteins, for the gene specific suppression of mitochondrial gene transcripts.
The study is well designed, conclusions are well supported.
The authors could have added more phenotypic information with regards to the flowering and anther/pollen formation of these plants.
Author Response
The authors could have added more phenotypic information with regards to the flowering and anther/pollen formation of these plants.
Response: quantitative data on the growth of the transformants at 2 and 3 weeks have been added in Figure 5b, but we do not have data on flowering.
Reviewer 2 Report
This study on Pentatricopeptide repeat (PPR) proteins and their potential use in manipulating organelle gene expression in plants is quite intriguing. The authors have done a good job in explaining the importance of PPR proteins and their ability to recognize different RNA sequences. They have also provided insights into the molecular basis of this sequence recognition.
The creation of a library of redesigned PPR proteins and their transformation into plants to target mitochondrial transcripts is a novel approach. The fact that ninety different variants were tested in vivo and showed a wide range of phenotypes is commendable. This indicates that there is potential for using PPR proteins as a tool for manipulating gene expression in plants.
The specific example of a line with slow growth and downward curled leaves, which showed a reduction in complex V due to specific cleavage of atp1 transcripts induced by a modified PPR protein, is a strong evidence of the effectiveness of the library. This validation of the library as a source of mitochondrial 'mutants' is significant.
Overall, this study is a step towards developing specific RNA targeting tools using PPR proteins that can be aimed at desired targets. It opens up new possibilities in plant biotechnology and genetic engineering. The findings of this study have implications in agricultural research and could potentially lead to the development of crops with improved traits.
However, there are a few areas where the authors could improve. Firstly, the study could benefit from more detailed explanations of the methods used in creating the library of redesigned PPR proteins. Secondly, it would be helpful to provide more information on the phenotypes observed in the other lines tested, in addition to the one mentioned in the study. This would give a more comprehensive understanding of the range of effects that can be achieved using the library.
In conclusion, this study on PPR proteins and their potential use in manipulating organelle gene expression in plants is a valuable contribution to the field. The authors have presented compelling evidence of the effectiveness of the library of redesigned PPR proteins. With some minor improvements, this study could have an even greater impact in the field of plant biotechnology.
Author Response
However, there are a few areas where the authors could improve. Firstly, the study could benefit from more detailed explanations of the methods used in creating the library of redesigned PPR proteins. Secondly, it would be helpful to provide more information on the phenotypes observed in the other lines tested, in addition to the one mentioned in the study. This would give a more comprehensive understanding of the range of effects that can be achieved using the library.
Response: additional details about the design of the gBlocks used for creating the library were added to the Methods section, lines 298-301: “To generate the PPR library, variant RPF2 motifs were designed as gBlocks Gene Fragments (GenScript), which can vary between 251 and 500 bp in length, with 18 potential variable bases in the central part, and a minimum of 125 bp fixed sequence at each end”.
Quantitative data on the growth of the transformant lines at 2 and 3 weeks have been added and show that about a quarter of them are growing much more slowly than the native RPF2 (WT+RPF2) control (Figure 5b).
Reviewer 3 Report
1. This study uses a standard system of PPR proteins to modify phenotypes of Arabidopsis by altering the pattern of mitochondrial transcript expression. The results showed some interesting findings but the experiential design has weak points mainly due to the lacking of replicates together with proper statistics to evaluate significant effects of treatments or phenotypic alterations or expression pattern changes by collecting quantitative data.
2. Large part of the experiments focus on one variant showed a reduction in complex V and the target of atp1, so the manuscript title should be more concentrated, informative, and specific.
3. Keywords: It's better to add additional 2-3 key terms.
4. Figure 2: A similar problem as mentioned in point 1, lacks quantitative data for proving the slow-growth phenotype of the variant.
5. Figures 6 and 7: Lacks quantitative data.
6. Overall quality: In my opinion, this work is a continuation of previous findings but looks preliminary and needs more evidence particularly collecting sufficient quantitative data to enrich it.
Author Response
- This study uses a standard system of PPR proteins to modify phenotypes of Arabidopsis by altering the pattern of mitochondrial transcript expression. The results showed some interesting findings but the experiential design has weak points mainly due to the lacking of replicates together with proper statistics to evaluate significant effects of treatments or phenotypic alterations or expression pattern changes by collecting quantitative data.
Response: Replicates were performed for all experiments, and we are sorry if it was not clear enough in the manuscript. Additional information has been added throughout the manuscript, in particular the analysis of growth rate in the transformants.
- Large part of the experiments focus on one variant showed a reduction in complex V and the target of atp1, so the manuscript title should be more concentrated, informative, and specific.
Response: The aim of this study is to create a library of PPR variants that will affect the expression of mitochondrial transcripts. We therefore feel that the title represents well this aim and should not be focused on one variant.
- Keywords: It's better to add additional 2-3 key terms.
Response: 2 key words were added: ATP synthase, RNA processing factor 2 (RPF2)
- 4. Figure 2: A similar problem as mentioned in point 1, lacks quantitative data for proving the slow-growth phenotype of the variant.
Response: Quantitative data based on image analysis showing the slow growth of the #280 and other transformants from the library have been provided in Figure 5.
- Figures 6 and 7: Lacks quantitative data.
Response: The analysis for the growth of the #280 seedlings is now provided in Figure 5b.
For Figure 7, RT qPCR was performed on 3 independent #280 plants. The phrase: “The error bars represent the standard deviation between 3 biological replicates (3 technical replicates each)“ was added to the legend.
- Overall quality: In my opinion, this work is a continuation of previous findings but looks preliminary and needs more evidence particularly collecting sufficient quantitative data to enrich it.
Response: as stated above, a quantitative analysis of the growth of the transformants from the library has been added.
Round 2
Reviewer 3 Report
The manuscript has been revised and I don't have further questions.